# Quest of Intelligent Research Tools for Rapid Evaluation of Fish Quality: FTIR Spectroscopy and Multispectral Imaging Versus Microbiological Analysis

**DOI:** 10.3390/foods10020264

**Published:** 2021-01-28

**Authors:** Maria Govari, Paschalitsa Tryfinopoulou, Foteini F. Parlapani, Ioannis S. Boziaris, Efstathios Z. Panagou, George-John E. Nychas

**Affiliations:** 1Laboratory of Microbiology and Biotechnology of Foods, Department of Food Science and Human Nutrition, School of Food and Nutritional Sciences, Agricultural University of Athens, Iera Odos 75, 11855 Athens, Greece; margochem@gmail.com (M.G.); ptryf@aua.gr (P.T.); stathispanagou@aua.gr (E.Z.P.); 2Laboratory of Marketing and Technology of Aquatic Products and Foods, Department of Ichthyology and Aquatic Environment, School of Agricultural Sciences, University of Thessaly, Phytoko Street, 38446 Volos, Greece; fwparlap@uth.gr (F.F.P.); boziaris@uth.gr (I.S.B.)

**Keywords:** sea bass fillets, FTIR spectroscopy, multispectral imaging, modified atmosphere packaging, PLS-R

## Abstract

The aim of the present study was to assess the microbiological quality of farmed sea bass (*Dicentrarchus labrax*) fillets stored under aerobic conditions and modified atmosphere packaging (MAP) (31% CO_2_, 23% O_2_, 46% Ν_2_,) at 0, 4, 8, and 12 °C using Fourier transform infrared (FTIR) spectroscopy and multispectral imaging (MSI) in tandem with data analytics, taking into account the results of conventional microbiological analysis. Fish samples were subjected to microbiological analysis (total viable counts (TVC), *Pseudomonas* spp., H_2_S producing bacteria, *Brochothrix thermosphacta*, lactic acid bacteria (LAB), Enterobacteriaceae, and yeasts) and sensory evaluation, together with FTIR and MSI spectral data acquisition. *Pseudomonas* spp. and H_2_S-producing bacteria were enumerated at higher population levels compared to other microorganisms, regardless of storage temperature and packaging condition. The developed partial least squares regression (PLS-R) models based on the FTIR spectra of fish stored aerobically and under MAP exhibited satisfactory performance in the estimation of TVC, with coefficients of determination (R^2^) at 0.78 and 0.99, respectively. In contrast, the performances of PLS-R models based on MSI spectral data were less accurate, with R^2^ values of 0.44 and 0.62 for fish samples stored aerobically and under MAP, respectively. FTIR spectroscopy is a promising tool to assess the microbiological quality of sea bass fillets stored in air and under MAP that could be effectively employed in the future as an alternative method to conventional microbiological analysis.

## 1. Introduction

Aquaculture is the most promising solution for the provision of animal protein for the increasing world population and the reduction of international poverty [1,2]. Nevertheless, a significant percentage of the total fish production (approximately 35%) is lost along the food supply chain due to mechanical damage, microbial growth, and/or microbial contamination, threatening food security and sustainability [3]. Of this quantity of fish production, 27% is wasted in post-farm gate practices, such as handling, storage, etc. [3]. Microbial activity is the most important cause of fish discards in post-harvest handling, storage, processing, and distribution (post-harvest losses). Under particular conditions in which fish are exposed after catch (e.g., experiencing temperature and time effects), bacteria can grow and produce metabolites that result in quality loss and the sensory rejection of the product [4]. Additionally, enzymatic activity and chemical oxidations can also occur, which are responsible for the production of spoilage-related compounds in fish tissue [4].

The most common preservation method used to retard the deterioration of fresh fish is refrigeration under aerobic conditions. However, the presence of oxygen at levels of 21% can favor the growth and metabolic activity of the most important microbial spoilers of fish such as *Pseudomonas* spp., and *Psychrobacter* spp. [4]. On the other hand, packaging in atmospheric conditions enriched in CO_2_ (modified atmosphere packaging (MAP)) has been recognized as an effective method for food preservation, including for seafood such as the European sea bass [5]. In MAP, the use of oxygen and carbon dioxide at different concentration levels can inhibit the growth of such spoilers compared to aerobic conditions, thus significantly extending (by 20–100%) the shelf life of the product [5,6].

The evaluation of spoilage status is important in determining the remaining shelf life of fish. It has been reported that fish usually spoils when its total microbial population exceeds 10^7^ CFU/g [4,7]. Bacterial enumeration based on agar culture media is reliable, but is also laborious and time consuming. Meanwhile, the fish industry and food authorities require fast, accurate and cost-effective methods to monitor microbial population and evaluate the microbiological quality of fish. To meet stakeholders’ demands, food microbiologists must develop intelligent research-led approaches and establish rapid, reliable and easy-to-use methodologies and technologies for the evaluation of fish quality in an extremely short time. This scenario will minimize food and economic losses for the aquaculture sector, fish processors, and distributors, thereby contributing to sustainable development goals [3].

In recent years, several hyperspectral and vibrational spectroscopy techniques have been developed to evaluate chemical composition and microbial spoilage in foods [8,9,10,11]. Fourier transform infrared (FTIR) spectroscopy, when combined with partial least squares regression (PLS-R) analysis, has been reported as a promising analytical method for the detection and quantification of spoilage bacteria in meat [12,13] and, recently, in fish [14]. Multispectral imaging (MSI), a promising rapid and non-invasive technology, obtains spatial and spectral information to evaluate the food spoilage and quality characteristics of food, including fish [15,16,17].

European sea bass (*Dicentrarchus labrax*) is one of the most important fish species farmed in Mediterranean countries. In 2019, the European Union was the largest producer of sea bass in the world, providing 80% of the annual world production, while Greece produced just over one half (51%) of the total EU production of European sea bass [18]. Apart from whole or gutted fish, fresh sea bass is also distributed in the EU market as fillets, stored either aerobically or under MAP in refrigerated conditions. The aim of the present study was to investigate the potential of FTIR spectroscopy and MSI as means of evaluating the microbiological quality of sea bass fillets packaged aerobically and under MAP at different isothermal conditions.

## 2. Materials and Methods

### 2.1. Fish Fillet Samples, Storage Conditions and Sampling

Farmed sea bass (*Dicentrarchus labrax*) fillets (about 250 g each) were obtained from Selonda Aquaculture S.A. (Athens, Greece) and transported to the laboratory on ice 48 h after harvesting. The sea bass fillets were supplied in packs in air or under modified atmosphere packaging (MAP) conditions (31% CO_2_, 23% O_2_, 46% N_2_). Upon arrival, the packages were stored at isothermal conditions (0, 4, 8, and 12 °C) in high precision (±0.5 °C) incubators (MIR-153, Sanyo Electric Co., Osaka, Japan). The objective was to simulate the available temperatures in retail fish outlets. The temperature profile of incubators was recorded at 15-min intervals throughout the experiment by means of electronic temperature-monitoring devices (COX TRACER, Cox Technologies Inc., Belmont, NC, USA).

Sea bass fillets were analyzed upon arrival to the laboratory (*t* = 0) and at regular time intervals depending on storage temperature. Specifically, fish fillet samples in both packaging trials were analyzed at 24, 12, 8, and 4–6 h during storage at 0, 4, 8, and 12 °C, respectively. Sea bass fillets under aerobic packaging conditions were stored at 0 and 4 °C for 287 and 143 h, respectively, whereas other fillets were stored at 8 and 12 °C for 94 h. Moreover, sea bass fillets under MAP conditions were stored at 0, 4, 8, and 12 °C for 545, 377, 281, and 209 h, respectively. For each sampling point, duplicate sea bass fillets from randomly selected packages under aerobic or MAP conditions at each storage temperature were subjected to: (i) microbiological analysis and pH measurement, (ii) sensory analysis, (iii) FTIR spectroscopy measurements, and (iv) MSI spectra acquisition.

### 2.2. Microbiological Analysis

For microbiological analysis, 25 g fish fillets were transferred aseptically in a stomacher bag (Seward Medical, London, UK), containing 225 mL of sterilized peptone saline diluent (0.1%, *w*/*v*, peptone, Neogen, Lansing, MI, USA) and 0.85%, *w*/*v*, sodium chloride, (Radiova, Praha, Slovakia), and homogenized in a Stomacher device (Lab Blender 400, Seward Medical, London, UK) for 60 s at room temperature. Serial 10-fold dilutions were prepared with the same diluent, and duplicate 0.1 mL portions of the appropriate dilutions were spread on the following media: Plate count agar (Biolife, Milano, Italy) for the enumeration of total viable counts (TVC), incubated at 25 °C for 3 d; *pseudomonas* agar base supplemented with Cephaloridine Fusidin Cetrimide (Neogen, Lansing, MI, USA) for *Pseudomonas* spp., incubated at 25 °C for 2 d; streptomycin sulphatethallous acetate cycloheximide (actidione) agar (Biolife, Milano, Italy) Milano, Italy for *B. thermosphacta*, incubated at 25 °C for 3 d; rose bengal chloramphenicol agar (Lab M, Lancashire, UK) for the enumeration of yeasts, incubated at 25 °C for 5 d. Similarly, duplicate 1.0 mL portions of the appropriate dilutions were mixed with the following media: de Man–Rogosa–Sharpe agar (Biolife, Milano, Italy) for lactic acid bacteria (LAB) incubated at 25 °C for 3–5 d; violet red bile glucose (Biolife, Milano, Italy) agar for the enumeration of Enterobacteriaceae, incubated at 37 °C for 24 h; iron agar (IA) for the enumeration of H_2_S producing bacteria by counting black colonies, after incubation at 25 °C for 3 d, according to Gram et al. [19]. After solidification of the inoculated media, a fold of 10 mL of the corresponding melting medium was added to cover the surface. Microbial characterization was confirmed by using microscopy observation, Gram staining, oxidase and catalase results.

Growth data of the different microorganisms enumerated on fish fillets were further modeled as a function of time, using the primary model of Baranyi and Roberts [20] to estimate the kinetic parameters. For data fitting, the program DMFit was used (www.combace.cc).

### 2.3. Gas Analysis

The gas composition (CO_2_ and O_2_) of the fish fillets packages was analyzed using a headspace gas analyzer Dansensor Chekmate 9900 (Tendringpacific, Denmark).

### 2.4. Sensory Analysis

The sensory attributes of odor and skin color of fish fillets were assessed in duplicate. Each attribute was scored on a five-point hedonic scale. A score of one to two corresponded to high-quality fillets without any off-odors and a bright skin color (fresh). A score of three characterized fish fillets with slight-off odors or without intense skin color but of acceptable quality (intermediate freshness), whereas a score of four to five corresponded to fish fillets of unacceptable quality. Scores exceeding the value of three indicated the end of a fish fillet’s shelf life.

### 2.5. FTIR Spectroscopy

The sea bass fillets (skin) were analyzed by FTIR spectroscopy in parallel to the microbiological analysis. A ZnSe 45° horizontal attenuated total reflectance (HATR) crystal (PIKE Technologies, Madison, WI, USA), an FTIR-6200 JASCO spectrometer (Jasco Corp., Tokyo, Japan) equipped with a triglycine sulphate detector, and a Ge/KBr beam splitter were used for to collect the FTIR spectral data of the sea bass fillets. The Spectra Manager Code of Federal Regulations (CFR) software version 2 (Jasco Corp.) was used, in accordance with Fengou et al. [14]. The FTIR spectra were further analyzed in the approximate wave number ranges of 3100 to 2700 and 1800 to 900 cm^−1^, as described previously for TVC [14].

### 2.6. Image Acquisition

Multispectral images of the sea bass fillet samples (skin) were obtained using the Videometer lab device (Videometer A/S, Hørsholm, Danemark), originally developed by the Technical University of Denmark. The Videometer lab device operates in reflectance mode from the ultraviolet (UV, 405 nm) to shortwave near-infrared (NIR, 970 nm) regions. The operation used for image acquisition and model development has been described in detail in previous works [14,21,22].

### 2.7. Data Analysis

Spectral Data Analysis and Correlation with Microbiological Data

Partial least squares regression (PLS-R) models were developed for the quantitative analysis of the microbial population of TVC using the spectral information of FTIR and MSI as input variables and the counts of TVC as output variables. The developed PLS-R models were evaluated to provide predictions in a temperature-independent manner, as previously described [14,21]. Model calibration was based on the data derived from fish fillet samples stored at 0 and 4 °C (*n* = 144), whereas model validation was undertaken using spectral data obtained during fish fillet storage at 8 and 12 °C (*n* = 74). Multivariate data analysis was carried out using The Unscrambler ver. 9.7 software (CAMO Software AS, Oslo, Norway). The performance of the developed PLS-R models was evaluated by the calculation of the following parameters: slope, offset, root mean squared error of calibration (RMSEc), cross-validation (RMSEcv), and prediction (RMSEp), as well as the coefficient of determination (R^2^) for calibration, cross-validation, and prediction. The optimum number of latent variables (LVs) was assigned at the minimum prediction residual error sum of squares after cross-validation using the leave-one-out cross-validation (LOOCV) method. Prior to analysis, spectral data were pre-processed by baseline correction.

## 3. Results and Discussion

### 3.1. Microbiological Spoilage of Sea Bass Fillets

The initial TVC of the sea bass fillets was 4.94 log CFU/g (Figure 1), which was about 1–2 logs higher compared to whole unprocessed or gutted fish [5,6,23]. This difference could be attributed to the use of equipment, utensils and the handling of fish fillets on working surfaces that could result in contamination, even in an industrial environment with good hygiene and sanitary practices.

Fish spoilage can usually occur when the TVC or Specific Spoilage Organisms (SSOs) reach around 7.0–8.0 log CFU/g. At these population levels, the activity of the microorganisms results in the production of metabolic compounds that contribute to the organoleptic rejection of the product [4,7]. Based on the sensory assessment, in aerobic storage conditions, the product was characterized as unacceptable after 191, 125, 78, and 48 h when at 0, 4, 8, and 12 °C, respectively, while it was characterized as unacceptable in MAP storage conditions after 305, 185, 88, and 71 h, at the same temperatures. Indeed, at the time of sensory rejection, TVC ranged between 9.0–9.8 log CFU/g for fish fillets stored aerobically and 7.0–8.0 log CFU/g for fish fillets under MAP storage conditions.

*Pseudomonas* spp. and H_2_S-producing bacteria, primarily *Shewanella* spp. [24], were found at higher population levels compared to the other studied microorganisms (Enterobacteriaceae, *B. thermosphacta*, yeasts, and LAB). *Pseudomonas* spp. grew faster in aerobic conditions, while its growth was inhibited under MAP [25,26]. In our study, the concentration of CO_2_ (31%) in combination with the concentration of O_2_ (23%) resulted in the outgrowth of pseudomonads and their dominance at all storage temperatures both in air and MAP conditions. This is also supported by the lower growth rates estimated in MAP compared to those of aerobic storage (Table 1). Indeed, *Shewanella* spp. presented the highest growth rate, despite the lower initial population (3.02 and 3.68 log CFU/g), in comparison to pseudomonads under all storage conditions (Table 1). Enterobacteriaceae reached the same final population, not exceeding 6.5–7.5 and 8.0–8.7 log CFU/g at 0, 4 and 8, 12 °C, respectively, regardless of packaging (Figure 1 and Figure 2). The MAP conditions at the higher temperatures (8 and 12 °C) favored the growth of *B. thermosphacta*, LAB, and yeasts, all of which were enumerated at least 1.8–2.0 log CFU/g higher than the corresponding populations in aerobic conditions.

Lerfall et al. [27] evaluated the growth changes of pseudomonads at 4 °C in different modified atmospheres in saithe and reported a growth of 3 logs in low CO_2_/high O_2_ fish samples (31.3% CO_2_/66.0% O_2_/2.7% N_2_) only, whereas in the other gas combinations assayed, no growth could be detected. Koutsoumanis et al. [25] studied the combined effect of storage temperature and packaging conditions (air and MAP) on red mullet and reported that the main spoilage bacteria in air were *Pseudomonas* spp. and *Shewanella* spp., whereas in MAP conditions *Shewanella* spp. and other facultative anaerobes dominated. In sea bass fish and sea bass fillets, H_2_S producing bacteria were found at similarly higher population levels compared to LAB and *Br. thermosphacta* under MAP storage conditions [5,6,28].

### 3.2. MAP Gas Analysis

The changes in gas concentration of MAP conditions during storage are presented in Figure 3. The concentration of gases remained almost unchanged until the time of sensory rejection, regardless of storage temperature. However, after the point of sensory rejection, CO_2_ presented a gradual increase, whereas O_2_ decreased until the end of storage. This was more pronounced at the higher storage temperatures (8 and 12 °C).

### 3.3. Sensory Analysis

Sensory evaluation (odor and skin color) of the raw fish fillets was also performed. Both attributes presented a decreasing trend throughout storage (Figure 4). It must be underlined that sensory rejection coincided with the time when TVC ranged from 6.7 to 7.7 log CFU/g for fish fillets stored under MAP conditions and the time when TVC was greater than 9.0 log CFU/g for aerobically stored fish fillets. Kritikos et al. [6] investigated the microbial spoilage of sea bass fillets in comparison with their volatilome and reported that the time of sensory rejection for cooked fish coincided with TVC level, ranging from 6.5 to 7.0 log CFU/g. In addition, Parlapani et al. [5] reported a good association between TVC and sensory rejection time for whole gutted sea bass fish stored at 2 °C.

### 3.4. Fish Spoilage Assessment Using FTIR and MSI Spectral Data

Examples of typical FTIR spectra of fresh (TVC of 4.9 log CFU/g) and spoiled (TVC of 8.50 log CFU/g) sea bass fillets stored under air and MAP are shown in Figure 5. The sensory scores for spoiled sea bass fillets corresponded to those of fish samples stored at 8 °C for 144 h and 12 °C for 88 h for the air and MAP trials, respectively. The FTIR spectra in the ranges of 3100–2700 and 1800–900 cm^−1^ provide an overall fingerprint of the biochemical composition and freshness of the fish. The peak at 1640 cm^−1^ (O–H stretch) is mostly due to water and amide I. The peak at 1545 cm^−1^ (N–H bend, C–N stretch) is ascribed to amide II. The peaks at 1314 and 1238 cm^−1^(C–N stretch, N–H bend, C=O–N bend) are related to amide III. The peaks at 1162–1025 cm^−1^ could be due to amines (C–N stretch) [14]. Most of these peaks correspond to amides and amines that could be mainly attributed to the microbial proteolytic activity occurring during storage [14]. The growth of *Pseudomonas* spp. in fish is related with the production of aldehydes, ketones, and ethyl esters (such as ethyl isovalerate or ethyl tiglate), which are considered fish spoilage components [29]. The development of ethyl esters and carbonyls in fish has been associated with peaks in the spectral regions of 2995–2860 and 1750–1705 cm^−1^, respectively [14]. Since *Pseudomonas* spp. was the dominant microbial group in the present study for fish fillets stored aerobically or under MAP, the aforementioned increased absorption peaks observed for the sea bass fillets (Figure 3) could be indicative of the production of spoilage metabolites, such as ethyl esters [14,28].

The performance of the developed PLS-R models based on the FTIR data for the prediction of TVC of sea bass fillets stored in air and MAP is shown in Figure 6, while the performance metrics are summarized in Table 2. Specifically, for the sea bass fillets stored under aerobic conditions, the coefficients of determination (R^2^) were 0.73, 0.68, and 0.78 for model calibration, cross-validation and prediction, respectively. Similarly, for the sea bass fillets stored under MAP, the coefficients of determination (R^2^) were 0.99, 0.72, and 0.99 for model calibration, cross-validation and prediction, respectively. The values of RMSE of prediction were low (0.84 and 0.64 for the sea bass fillets stored aerobically and under MAP, respectively). According to Chin [30], R^2^ values higher than 0.67 indicate a high predictive accuracy of the PLS-R model; ranges between 0.33–0.67 and 0.19–0.33 indicate moderate and low accuracy, respectively, while values below 0.19 are considered unacceptable. High R^2^ values and low RMSE values of calibration, cross-validation, and prediction indicate the good accuracy and precision of PLS-R models [31]. Therefore, the developed PLS-R models indicated satisfactory performance between spectral and TVC data that could provide reliable predictions of microbial populations regardless of storage temperature.

There is a limited number of publications for the estimation of microbial counts of fish directly from FTIR spectral data. Specifically, Saraiva et al. [32] investigated the potential of FTIR to predict the bacterial load of salmon fillets (*Salmo salar*) stored at 3, 8, and 30 °C in aerobic and MAP conditions. The developed PLS-R model for the prediction of TVC showed R^2^ and RMSE prediction values of 0.81 and 0.78, respectively. The authors concluded that FTIR can be used as a reliable, accurate, and fast method for real time freshness evaluation of salmon fillets stored under refrigerated conditions. In a recent study [14], FTIR spectroscopy was employed in tandem with multivariate data analysis [33] for the assessment of the microbiological quality of farmed whole ungutted sea bream (*Sparus aurata*) stored aerobically at 0, 4, and 8 °C. In this study, the PLS-R model developed on spectral data obtained from the fish skin provided a satisfactory prediction of TVC, as inferred by the values of the coefficient of determination (R^2^, 0.72) and the root mean square error (RMSE, 0.71). In a recent work on grass carp (freshwater fish) [34], a rapid, online multispectral imaging system was employed to predict freshness based on chemical indices with satisfactory results. In a different approach, FTIR was used to detect the adulteration between two fish species, namely salmon and salmon trout [35]. The authors concluded that FTIR combined with different data analysis techniques (such as PCA and PLS-R) could effectively predict the presence/absence of adulteration in fish samples.

The performance of the developed PLS-R models based on MSI spectral data for the estimation of TVC of sea bass fillets stored under aerobic conditions and MAP is presented in Figure 7, while the performance metrics are shown in Table 3.

In general, the performances of the PLS-R models based on MSI spectral data were less satisfactory compared to those of models based on FTIR. Specifically, aerobically stored sea bass fillets presented lower values for the coefficient of determination (R^2^) (namely 0.58, 0.48, and 0.43) for model calibration, cross-validation, and prediction, respectively. Similarly, for the sea bass fillets stored under MAP, the coefficients of determination (R^2^) of the developed PLS-R model were 0.65, 0.53, and 0.62 for model calibration, cross-validation, and prediction, respectively. In addition, the values of RMSE of prediction were 1.31 and 0.76 for the sea bass fillets stored under aerobic conditions and stored under MAP, respectively. In agreement with the present study, a previously published PLS-R model based on MSI spectral data acquired from the skin and flesh of sea bream did not perform satisfactorily to assess the microbiological quality of fish regarding TVC prediction [14]. However, Cheng and Sun [10] used visible and near-infrared (Vis-NIR) hyperspectral imaging in the range of 400–1000 nm for the prediction of TVC on the flesh of grass carp during refrigerated storage at 4 °C and reported that the developed PLS-R model based on the full wavelength data provided a satisfactory performance in the prediction of TVC in terms of RMSE (0.57) and R^2^ (0.90) values. Therefore, the MSI spectral changes of microbial growth may be different for various fish species.

## 4. Conclusions

FTIR spectroscopy is a promising approach to predict the microbiological quality of sea bass fillets stored aerobically and under MAP when compared with conventional microbiological analysis. In contrast, MSI presented a less satisfactory performance and could not be used effectively for quality assessment of sea bass fillets. Our findings will be used to develop a rapid, reliable, and easy-to-use toolkit to evaluate the microbiological quality of sea bass fish fillets so that food microbiologists and seafood specialists can rapidly tackle the seafood quality-related concerns of stakeholders. Such innovations and improvements in food microbiology will minimize food losses, contributing to enhanced food security, welfare, and sustainability.

## Figures and Tables

**Figure 1 foods-10-00264-f001:**
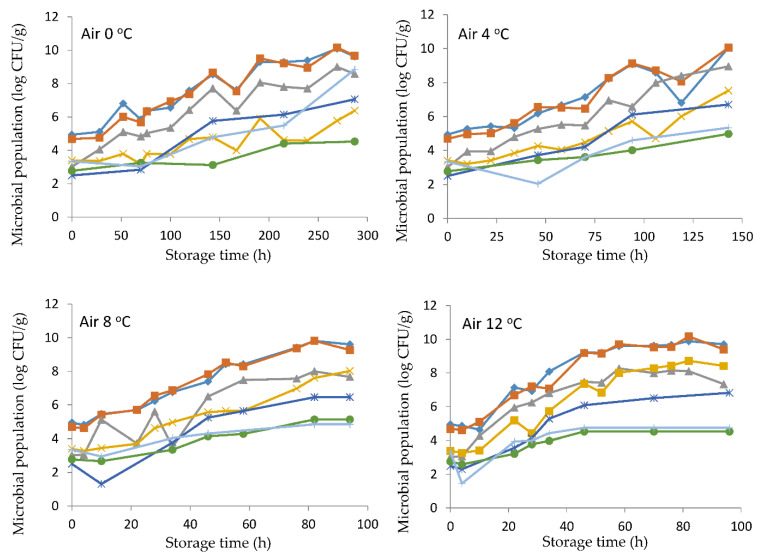
Growth of microorganisms on sea bass fish fillets at different temperatures (0, 4, 8, and 12 °C) in aerobic packaging conditions. Total viable counts: (◊), *Pseudomonas* spp.: (▪), H_2_S-producing bacteria: (∆), Enterobacteriaceae: (x), *B. thermosphacta*: (ж), yeasts: (**+**), and lactic acid bacteria: (●).

**Figure 2 foods-10-00264-f002:**
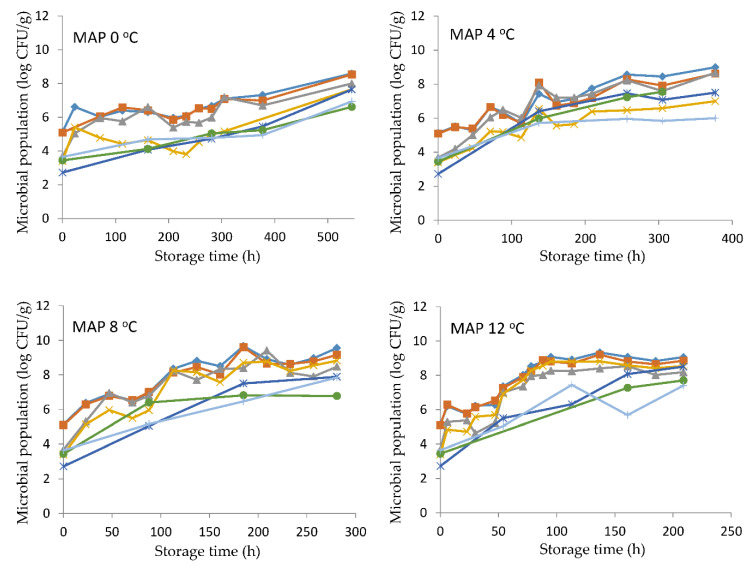
Growth of microorganisms on sea bass fish fillets stored at different temperatures (0, 4, 8, and 12 °C) and modified atmosphere packaging conditions. Total viable counts: (◊), *Pseudomonas* spp.: (▪), H_2_S-producing bacteria: (∆), Enterobacteriaceae: (x), *B. thermosphacta*: (ж), yeasts: (+), and lactic acid bacteria (●).

**Figure 3 foods-10-00264-f003:**
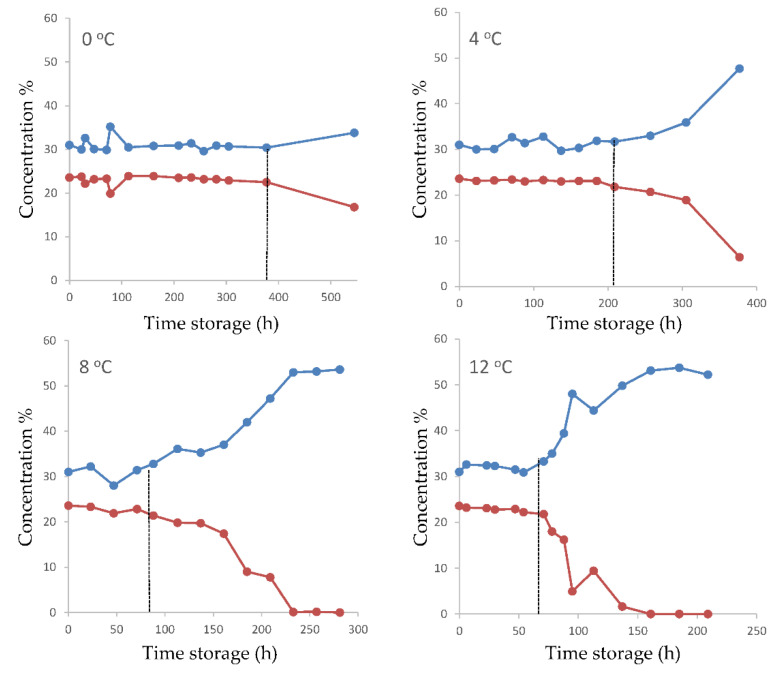
Changes in headspace concentration (%) of CO_2_ (∙) and O_2_ (∙) during fish fillet storage under modified atmosphere packaging at different temperatures. Dashed lines indicate the time of sample rejection.

**Figure 4 foods-10-00264-f004:**
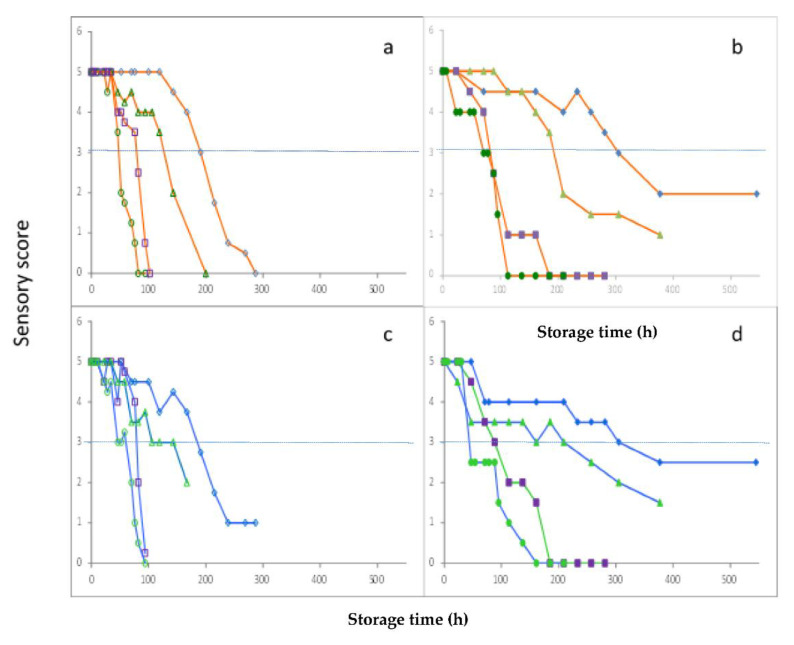
Sensory score for odor (**a**,**b**) and skin color (**c**,**d**) degradation of sea bass fish fillets stored in air (**a**,**c**) and MAP (**b**,**d**) at 0 °C (♦, ◊), 4 °C (▲, ∆), 8 °C (■, □), and 12 °C (●, ○). Dashed lines indicate the level of sample rejection.

**Figure 5 foods-10-00264-f005:**
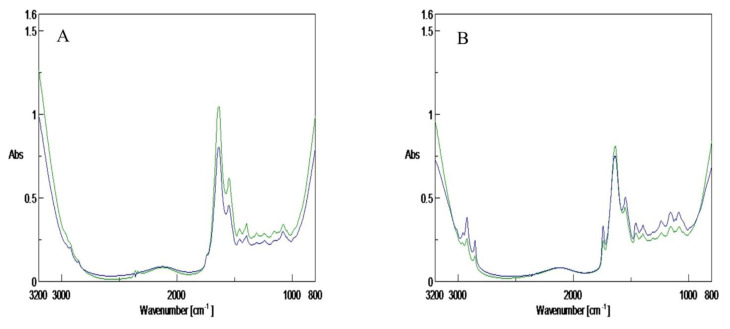
Representative FTIR spectra, in the wavenumber range of 3100 to 900 cm^−1^, corresponding to sea bass fillets stored in air (**A**) and MAP (**B**). Blue and green lines indicate fresh (total viable counts (TVC) around 4.9 log CFU/g) and spoiled (TVC around 8.5 log CFU/g) fish fillets, respectively.

**Figure 6 foods-10-00264-f006:**
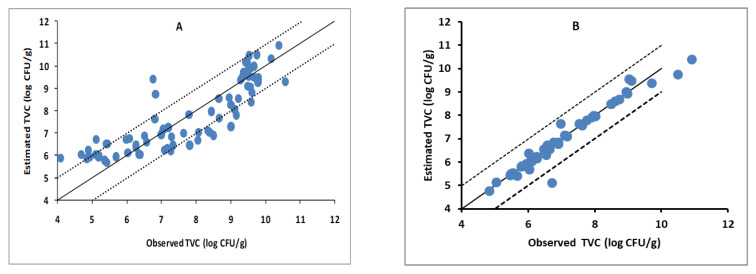
Comparison between observed and estimated counts of TVC by the PLS-R model based on FTIR spectral data for the sea bass fillets stored in air (**A**) and under MAP (**B**). Solid line indicates the line of equity (y = x); dashed lines indicate ±1 log unit area.

**Figure 7 foods-10-00264-f007:**
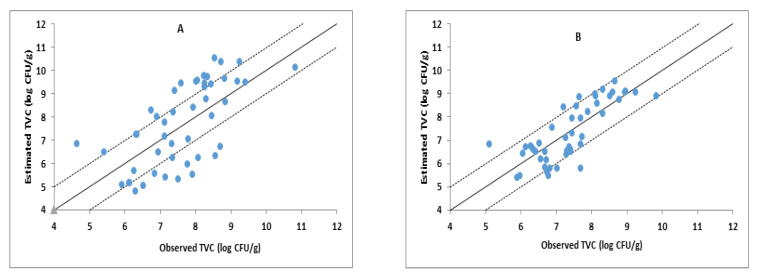
Comparison between observed and estimated counts of TVC by the PLS-R model based on MSI spectral data for the sea bass fillets stored in air (**A**) and under MAP (**B**).The solid line indicates the line of equity (y = x); dashed lines indicate ±1 log unit area.

**Table 1 foods-10-00264-t001:** Estimated kinetic parameters of different microbial groups in sea bass fish fillets stored aerobically and under modified atmosphere packaging (MAP) at different temperature conditions.

Temperature/Packaging		Air						MAP					
°C		y0	Rate	±SE	yEnd	Se Fit	R^2^	y0	Rate	±SE	yEnd	SE Fit	R^2^
	Total Viable Count	4.94	0.60	0.05	9.76	0.31	0.97	5.13	0.13	0.01	8.60 *	0.16	0.97
	*Pseudomonas* spp.	4.68	0.63	0.05	9.92	0.32	0.97	5.09	0.14	0.01	8.50 *	0.16	0.97
	*Shewanella* spp.	3.02	0.63	0.06	8.70	0.40	0.96	3.68	0.19	0.01	8.03	0.10	0.99
0	Enterobacteriaceae	3.40	0.25	0.02	6.37 *	0.23	0.95	3.94	0.25	0.03	7.70 *	0.21	0.96
	*B. thermophacta*	2.50	0.45	0.01	7.13	0.06	1.00	2.72	0.21	0.02	7.60 *	0.36	0.96
	Lactic acid bacteria	2.77	0.18	0.02	4.95 *	0.15	0.97	3.45	0.14	0.01	6.60 *	0.21	0.97
	Yeasts	3.34	0.70	0.05	8.85 *	0.19	0.99	3.65	0.10	0.01	6.90 *	0.13	0.97
	Total Viable Count	4.94	0.91	0.04	10.02 *	0.27	0.98	5.13	0.28	0.02	9.02	0.18	0.98
	Pseudomonas spp.	4.68	0.96	0.04	10.05 *	0.28	0.98	5.09	0.22	0.01	8.60 *	0.20	0.96
	*Shewanella* spp.	3.02	1.02	0.04	8.95 *	0.27	0.98	3.68	0.72	0.07	8.18	0.33	0.96
4	*Enterobacteriaceae*	3.40	0.79	0.09	7.52 *	0.22	0.97	3.40	0.32	0.03	6.73	0.26	0.94
	*B. thermophacta*	2.50	0.71	0.04	6.70 *	0.16	0.99	2.72	0.66	0.06	7.34	0.24	0.99
	Lactic acid bacteria	2.77	0.36	0.03	4.97 *	0.13	0.97	3.45	0.33	0.03	7.07 *	0.34	0.97
	Yeasts	3.34	1.08	0.03	5.34	0.01	1.00	3.65	0.39	0.02	5.90	0.06	1.00
	Total Viable Count	4.94	1.94	0.25	9.69	0.22	0.99	5.13	0.62	0.07	8.96	0.28	0.95
	*Pseudomonas* spp.	4.68	1.72	0.11	9.55	0.26	0.98	5.09	0.53	0.07	8.78	0.31	0.94
	*Shewanella* spp.	3.02	2.17	0.22	7.71	0.35	0.96	3.68	1.01	0.11	8.28	0.29	0.96
8	Enterobacteriaceae	3.40	1.25	0.06	8.00 *	0.27	0.97	3.40	0.90	0.10	8.60	0.35	0.96
	*B. thermophacta*	2.50	1.24	0.10	6.45 *	0.28	0.97	2.72	0.50	0.07	8.40 *	0.61	0.94
	Lactic acid bacteria	2.77	1.50	0.14	5.12	0.07	1.00	3.45	0.84	0.02	6.80	0.03	1.00
	Yeasts	3.34	0.46	0.02	4.84 *	0.07	0.99	3.65	0.36	0.01	7.80 *	0.11	1.00
	Total Viable Count	4.94	2.46	0.23	9.72	0.38	0.97	5.13	1.26	0.12	9.07	0.15	0.99
	*Pseudomonas* spp.	4.68	2.37	0.20	9.71	0.34	0.97	5.09	1.26	0.17	8.87	0.19	0.98
	*Shewanella* spp.	3.02	2.86	0.28	7.86	0.34	0.97	3.68	1.28	0.07	8.28	0.19	0.99
12	Enterobacteriaceae	3.40	2.03	0.15	8.56	0.33	0.97	3.40	1.47	0.08	8.68	0.22	0.99
	*B. thermophacta*	2.50	3.41	0.21	6.82 *	0.13	0.99	2.72	0.69	0.06	8.50 *	0.42	0.97
	Lactic acid bacteria	2.77	1.09	0.06	4.53 *	0.04	1.00	3.45	0.52	0.05	7.70 *	0.34	0.97
	Yeasts	3.34	0.73	0.06	4.75 *	0.09	0.97	3.65	0.43	0.04	7.40 *	0.29	0.96

y_0_ and y_End_: initial and final estimates (log CFU/g) of population; Rate: maximum specific growth rate (h^−1^); SE: standard error; * indicates no estimated values, only observed values.

**Table 2 foods-10-00264-t002:** Performance metrics of the partial least squares regression (PLS-R) models correlating TVC in sea bass fillet samples stored in air (A) and under MAP (B) on the basis of FTIR spectral data.

Storage	Data Set	Slope	Offset	R^2^	RMSE
Air	Calibration	0.73	2.01	0.73	0.90
	Cross-validation *	0.68	2.45	0.59	1.14
	Prediction	0.78	1.81	0.75	0.84
MAP	Calibration	0.96	0.28	0.96	0.67
	Cross-validation	0.72	5.41	0.99	1.05
	Prediction	0.99	0.24	0.99	0.64

R^2^: coefficient of determination; RMSE: root mean squared error; * Leave-one-out cross-validation.

**Table 3 foods-10-00264-t003:** Performance metrics of the PLS-R models correlating TVC in sea bass fillet samples stored in air (A) and under MAP (B) on the basis of multispectral imaging (MSI) spectral data.

Storage	Data Set	Slope	Offset	R^2^	RMSE
Air	Calibration	0.58	3.10	0.58	1.12
	Cross-validation *	0.48	3.89	0.40	1.38
	Prediction	0.43	4.23	0.44	1.31
MAP	Calibration	0.65	2.79	0.65	0.77
	Cross-validation	0.53	3.38	0.42	0.96
	Prediction	0.62	2.78	0.62	0.76

R^2^: coefficient of determination; RMSE: root mean squared error; * Leave-one-out cross-validation.

## Data Availability

The data presented in this study are available on request from the corresponding author. The data are not publicly available due to privacy.

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
