# Peer review of "Quest of Intelligent Research Tools for Rapid Evaluation of Fish Quality: FTIR Spectroscopy and Multispectral Imaging Versus Microbiological Analysis"

_foods, 2021, doi:10.3390/foods10020264_

Round 1

Reviewer 1 Report

The topic presented in this manuscript is very relevant in terms of human heath because addresses the big problem linked by consumption of contaminated food.  Luckily,  there are laws for their control worldwide in order to guarantee they quality both microbiological and chemical point of view.

The development  of innovative tools for the detection of contaminants is very relevant topic but in this manuscript is not easy to evaluate theirs effectiveness because the microbiological part is poor.

The microbiological analysis performed are not strictly written including the sampling. It needs for thorough revision. Along the manuscript the microbiological part must be re written because  relevant part are missing.

For example, which is the volume of sample analysed from each dilution?  Describe which technique was  used for  each target microorganism?  How the identification at species level was done?  Do you have performed the positive and negative control?

The results are expressed as CFU/g , do you have  done the dry weight ?

Further, the results interpretation is often inadequate for example

Line 166-167 pg 5 "The use of equipment, utensils and the 166 handling of fish fillets on working surfaces, even in an industrial environment with good hygiene 167
and sanitary practices, could result in cross-contamination between raw and final product..."

Again, the Fig. 1  is not clear  ,  the range  of assesses axis should be change to make more legible  the results.

The consideration on result interpretation are often not supported by result. The manuscripts needs of major revision.

Author Response

REVIEWER 1

The topic presented in this manuscript is very relevant in terms of human heath because addresses the big problem linked by consumption of contaminated food.  Luckily, there are laws for their control worldwide in order to guarantee they quality both microbiological and chemical point of view.

The development of innovative tools for the detection of contaminants is very relevant topic but in this manuscript is not easy to evaluate theirs effectiveness because the microbiological part is poor.

The microbiological analysis performed are not strictly written including the sampling. It needs for thorough revision. Along the manuscript the microbiological part must be re written because relevant part are missing.

For example, which is the volume of sample analysed from each dilution?  Describe which technique was used for each target microorganism? 

Response: We thank the reviewer for this comment that helps in the detailed description of the material and methods. The comment was adopted, and the text was modified accordingly (Rev. text line 115-127)

How the identification at species level was done?  Do you have performed the positive and negative control?

Response: the microorganisms were tested using microscopy observation, Gram staining, oxidase and catalase results.  

Your comment was adopted, and the text was modified accordingly (Rev. text line 127-128)

The results are expressed as CFU/g, do you have done the dry weight?

Response: We followed the standard microbiological technique (Harrigan and Mccance 1998, Laboratory methods in food microbiology)

Further, the results interpretation is often inadequate for example

Line 166-167 pg 5 "The use of equipment, utensils and the 166 handling of fish fillets on working surfaces, even in an industrial environment with good hygiene 167and sanitary practices, could result in cross-contamination between raw and final product..."

Response: This sentence is based on the fact that the initial low population of fresh un-gutted fish is gradually increased during processing treatments in the factory resulting in a higher final population of processed fish such as fillets (Rev. text line 173-175). The sentence was improved according to the comments of the second reviewer by adding a comma (,) after the word “utensils”.

Again, the Fig. 1 is not clear, the range of assesses axis should be change to make more legible  the results.

Response: Your comment was adopted and the Figure 1 was modified accordingly.

The consideration on result interpretation are often not supported by result. The manuscripts needs of major revision.

Response: A detailed revision of the manuscript has been undertaken based on the comments of both reviewers. Moreover, the results have been enhanced by the editing of the Figures in the manuscript, as recommended by the two reviewers.  

Reviewer 2 Report

Please standardize the abbreviations throughout the article; for example, MAP also appears as M.A.P.

Abstract

18 – with data analytics? Or data analysis?

information on the storage method is repeated on lines 17 and 19

44 – “which are” seem redundant

53 – extending thus > thus extanding

58 - quality  of fish – double spaced

60 – consider a change to: To meet stakeholders’ demands, food microbiologists attempt to develop intelligent research-led approaches and establish rapid, reliable, and easy-to-use methodologies and technologies to evaluate fish quality in an extremely short time

66 – for the evaluation of > to evaluate

70-71 – Consider change to: Multispectral imaging (MSI), a promising rapid and non-invasive technology, allows to obtain spatial and spectral information to evaluate food spoilage and quality characteristics of food, including fish [15-17]. 

74 - The European Union was the largest… if anything has changed, please explain or rephrase the sentence in a way that 2019 refers not only to the Greek share of production

73 - the paragraph from lines 73 to 78 does not match the third sentence of this paragraph. Please come to the scientific justification of the presented research in a logical manner. Divide the last sentence into two at least.

86 – in ice? Or on ice? Really? After 24 of harvesting?

87 - supplied in packs in air or under modified atmospherepackaging – please correct

89 - (0, 4, 8, and 12 oC) – please correct the degree sign

92 - the experiments > experiment

94 - for better clarity, prepare a visualization of the experiment pattern

101 – 102 - change to a capital letter ((i) etc.)

106 - > peptone, and 0.85%, w/v, NaCl

Please unify the description of the reagent / equipment manufacturer in accordance with the requirements of the publishing house

119 > modeled

120 > a function of time

121 – correct the sentence

123 - correct the sentence

128-130 > A score of 3 characterized fish fillets as slight off odors or without intense skin color but the acceptable quality (intermediate freshness), whereas a score of 4-5 corresponded to fish fillets of unacceptable quality.

135 > an FTIR-6200 JASCO

136 > triglycine sulfate

138 > was used, according to Fengou

145 > near-infrared; regions. The

151 > as the output variable

161 > validation using the leave-one-out cross-validation; analysis, spectral

165 > (Figure 1), which

166 > utensils, and

Figure 1 > improve charts, annotate axes, fit x-axis to each chart for better readability

177 reach at ca.

180 > conditions, the product

181 > 78, and 48 h; °C

187 > yeasts, and LAB

194 > °C

196 LAB, and yeasts

198 > aerobically, and

Figure 2 - improve charts, annotate axes, fit x-axis to each chart for better readability

223 > wordy sentence, please correct - However, after the point of sensory rejection, the concentration of CO2 presented a gradual increase, whereas the level of O2 decreased until the end of storage

Figure 3 > improve charts, annotate axes, fit x-axis to each chart for better readability

239 > reported a good

240 > 2 oC - did you put o in the superscript? please correct throughout the article

281-288 – add comas

303 > first case, the

304 > accurate, and; real-time

309 > provided a satisfactory

321> The solid

328 > cross-validation, and

331 > did not perform satisfactorily to assess the microbiological

332 > near-infrared

337 > for various fish species

339 > FTIR spectroscopy is a promising approach for predicting the microbiological quality of sea bass fillets stored aerobically and under MAP compared with microbiological analysis. In contrast, MSI presented less satisfactory performance and could not be used effectively for quality assessment of sea bass fillets. Our findings could be used to develop a rapid, reliable, and easy-to-use toolkit to evaluate the microbiological quality of sea bass fish fillets so that food microbiologists and seafood specialists tackle the seafood quality-related concerns of stakeholders rapidly. Such innovations and improvements in food microbiology will minimize food losses contributing to enhanced food security, welfare, and sustainability.

Please discuss the obtained results more broadly with the current literature

Author Response

REVIEWER 2

Please standardize the abbreviations throughout the article; for example, MAP also appears as M.A.P.

Response: The MAP was used as standard abbreviation throughout the article.   

Abstract

18 – with data analytics? Or data analysis?

Response: Data analytics; this is the term used nowadays

information on the storage method is repeated on lines 17 and 19

Response: Your comment was adopted, and the text was modified accordingly (Rev. text line 18-19)

44 – “which are” seem redundant

Response: Your comment was adopted, and the text was modified accordingly (Rev. text line 46)

53 – extending thus > thus extanding

Response: Your comment was adopted and the text was modified accordingly (Rev. text line 55)

58 - quality  of fish – double spaced

Response: Your comment was adopted, and the text was modified accordingly (Rev. text line 62)

60 – consider a change to: To meet stakeholders’ demands, food microbiologists attempt to develop intelligent research-led approaches and establish rapid, reliable, and easy-to-use methodologies and technologies to evaluate fish quality in an extremely short time

Response: Thank you for the comment, which was also adopted, and the text was modified accordingly (Rev. text line 62-66)

66 – for the evaluation of > to evaluate

Response: Your comment was adopted and the text was modified accordingly (Rev. text line 68)

70-71 – Consider change to: Multispectral imaging (MSI), a promising rapid and non-invasive technology, allows to obtain spatial and spectral information to evaluate food spoilage and quality characteristics of food, including fish [15-17]. 

Response: Thank you for the comment. It was adopted and the text was modified accordingly (Rev. text line 72-74)

74 - The European Union was the largest… if anything has changed, please explain or rephrase the sentence in a way that 2019 refers not only to the Greek share of production

Response: The text was improved and 2019 referred to both (Rev. text line 76-78)

73 - the paragraph from lines 73 to 78 does not match the third sentence of this paragraph. Please come to the scientific justification of the presented research in a logical manner. Divide the last sentence into two at least.

Response: The first part of the sentence has been deleted. Thank you for the comment. (Rev. text line 80-81)

86 – in ice? Or on ice? Really? After 24 of harvesting?

Response: This is corrected and the transporting time was 48 h based on the company (Rev. text line 88)

87 - supplied in packs in the air or under modified atmosphere packaging – please correct

Response: Your comment was adopted and the text was modified accordingly (Rev. text line 89)

89 - (0, 4, 8, and 12 oC) – please correct the degree sign

Response: Your comment was adopted and the symbol was corrected through the text.

92 - the experiments > experiment

Response: Your comment was adopted and the text was modified accordingly (Rev. text line 93)

94 - for better clarity, prepare a visualization of the experiment pattern

Response: This comment has been addressed by the preparation of a visualization figure which is provided as a graphical abstrct file.

101 – 102 - change to a capital letter ((i) etc.)

Response: Your comment was adopted and the text was modified accordingly (Rev. text line 107-108)

106 - > peptone, and 0.85%, w/v, NaCl

Please unify the description of the reagent / equipment manufacturer in accordance with the requirements of the publishing house

Response: Description of reagents and equipment are corrected according to the journal (Rev. text line 112-122)

119 > modeled

Response: Your comment was adopted and the text was modified accordingly (Rev. text line 125)

120 > a function of time

Response: Your comment was adopted, and the text was modified accordingly (Rev. text line 126)

121 – correct the sentence

Response: Your comment was adopted and the text was modified accordingly (Rev. text line 126-127)

123 - correct the sentence

Response: The sentence has been corrected. (Rev. text line 130-131)

128-130 > A score of 3 characterized fish fillets as slight off odors or without intense skin color but the acceptable quality (intermediate freshness), whereas a score of 4-5 corresponded to fish fillets of unacceptable quality

Response: Your comment was adopted and the text was modified accordingly (Rev. text line 136-137)

135 > an FTIR-6200 JASCO

Response: Your comment was adopted and the text was modified accordingly (Rev. text line 142)

136 > triglycine sulfate

Response: Your comment was adopted and the text was modified accordingly (Rev. text line 143)

138 > was used, according to Fengou

Response: Your comment was adopted and the text was modified accordingly (Rev. text line 146)

145 > near-infrared; regions. The

Response: Both of them are corrected (Rev. text line 152)

151 > as the output variable

Response: Your comment was adopted and the text was modified accordingly (Rev. text line 158)

161 > validation using the leave-one-out cross-validation; analysis, spectral

Response: Your comment was adopted and the text was modified accordingly (Rev. text line 168)

165 > (Figure 1), which

Response: Your comment was adopted and the text was modified accordingly (Rev. text line 172)

166 > utensils, and

Response: Your comment was adopted and the text was modified accordingly (Rev. text line 173)

Figure 1 > improve charts, annotate axes, fit x-axis to each chart for better readability

Response: Figure, and axes were modified accordingly.

177 reach at ca.

Response: Your comment was adopted and the text was modified accordingly (Rev. text line 196)

180 > conditions, the product

Response: Your comment was adopted and the text was modified accordingly (Rev. text line 199)

181 > 78, and 48 h; °C˚º˚

Response: Your comment was adopted and the text was modified accordingly (Rev. text line 200)

187 > yeasts, and LAB

Response: Your comment was adopted and the text was modified accordingly (Rev. text line 206)

194 > °C

Response: Your comment was adopted and the text was modified accordingly (Rev. text line 213)

196 LAB, and yeasts

Response: Your comment was adopted and the text was modified accordingly (Rev. text line 215)

198 > aerobically, and

Response: Your comment was adopted and the text was modified accordingly (Rev. text line 217)

Figure 2 - improve charts, annotate axes, fit x-axis to each chart for better readability

Response: Figure 2 revised accordingly

223 > wordy sentence, please correct - However, after the point of sensory rejection, the concentration of CO2 presented a gradual increase, whereas the level of O2 decreased until the end of storage

Response: Your comment was adopted and the text was modified accordingly (Rev. text line 259-260)

Figure 3 > improve charts, annotate axes, fit x-axis to each chart for better readability

Response: Your comment was adopted and the Figure 3 was modified accordingly

239 > reported a good

Response: Your comment was adopted, and the text was modified accordingly (Rev. text line 288)

240 > 2 oC - did you put o in the superscript? please correct throughout the article

Response: Your comment was adopted, and the symbol was corrected (Rev. text line 289)

281-288 – add comas

Response: Your comment was adopted, and the text was modified accordingly (Rev. text line 328)

303 > first case, the

Response: Your comment was adopted, and the text was modified accordingly (Rev. text line 360)

304 > accurate, and; real-time

Response: Your comment was adopted and the text was modified accordingly (Rev. text line 361)

309 > provided a satisfactory

Response: Your comment was adopted and the text was modified accordingly (Rev. text line 367)

321> The solid

Response: Your comment was adopted and the text was modified accordingly (Rev. text line 387)

328 > cross-validation, and

Response: Your comment was adopted and the text was modified accordingly (Rev. text line 391)

331 > did not perform satisfactorily to assess the microbiological

Response: Your comment was adopted and the text was modified accordingly (Rev. text line 397)

332 > near-infrared

Response: Your comment was adopted and the text was modified accordingly (Rev. text line 399)

337 > for various fish species

Response: Your comment was adopted and the text was modified accordingly (Rev. text line 403)

339 > FTIR spectroscopy is a promising approach for predicting the microbiological quality of sea bass fillets stored aerobically and under MAP compared with microbiological analysis. In contrast, MSI presented less satisfactory performance and could not be used effectively for quality assessment of sea bass fillets. Our findings could be used to develop a rapid, reliable, and easy-to-use toolkit to evaluate the microbiological quality of sea bass fish fillets so that food microbiologists and seafood specialists tackle the seafood quality-related concerns of stakeholders rapidly. Such innovations and improvements in food microbiology will minimize food losses contributing to enhanced food security, welfare, and sustainability.

Response: Your suggestion included to the text (Rev. text line 405-412)

 Please discuss the obtained results more broadly with the current literature

Response:  We do agree with the reviewer for this comment and we have added some additional text to discuss our results more broadly with reference to the current literature. Due to the diversity of fish composition and its effect on the performance of the used techniques, the discussion of results is focused on fish species of similar origin and composition. See revised manuscript, page 11, lines 368-376.

Round 2

Reviewer 1 Report

Accept

Author Response

thanks

Reviewer 2 Report

There are still a lot of linguistic errors throughout the article.

Please correct the reference in the added fragment.

Author Response

The authors thank the reviewer for his/her valuable suggestions;

The requested comments have been taken into account e.g.

(1) extensive editing and (2) improvement of Introduction

Once more thank you so much